# Analysis of Influencing Factors and Prevention of Coal Wall Deformation and Failure of Coal Wall in Caving Face with Large Mining Height: Case Study

**Guohao Meng [1], Jixiong Zhang [1,\*], Chongjing Wang [2], Nan Zhou [1] and Meng Li [1]**

[1]   State Key Laboratory of Coal Resources and Safe Mining, School of Mines,
     China University of Mining & Technology, Xuzhou 221116, China
[2]   Jining Energy Development Group Co., Ltd., Jining 272073, China
\*    Correspondence: cumtzjxiong@163.com; Tel.: +86-139-1200-5505

**Abstract:** The coal walls in a caving face with a tall mining height are prone to rib spalling, which leads to the phased cessation of the mining of the working face, causes heavy losses, and endangers the safety of underground workers. In order to prevent serious rib spalling accidents of coal walls in fully mechanized caving faces with a large mining height and to improve the prediction of and ability to control rib spalling, a load-bearing mechanical model of the roof–coal wall–support system was established based on the moment-balance relationship. The expressions for the deformation and stress distribution in a coal wall were calculated. Then, the influences of key factors on the horizontal displacement of the coal wall were investigated. A numerical simulation model of the working face was established, and an orthogonal test design was introduced. On this basis, the influences of four factors: cutting height, breaking position of the main roof, support strength, and sidewall protecting force of the support on the horizontal displacement and volume of a plastic zone of coal wall, were analyzed. Moreover, their order of importance was ranked on the basis of sensitivity. Based on the engineering conditions and production practices in the Cuncaota II Coal Mine, key parameters for controlling and measures for preventing the rib spalling of the coal wall are proposed to guide practical actions.

**Keywords:** coal mining; large mining height; rib spalling; sensitivity analysis

## 1. Introduction

Thick coal seams (thickness $\geq 3.5$ m) are widely distributed in China, especially in Inner Mongolia Autonomous Region, Shaanxi Province, Shanxi Province, and Xinjiang Uygur Autonomous Region. According to statistics, the output of the mining of thick coal seams accounts for more than 40% of the total coal output in China [1,2]. Fully mechanized top-coal caving technology in thick coal seams provides an important technical guarantee to improve the yield per unit of the working face. The Cuncaota II Coal Mine in the Shendong mining area is located in Ordos City, Inner Mongolia Autonomous Region, China. The #31 and #42 coal seams in the southwest of the mine field are combined and mined via a fully mechanized top-coal caving process. In recent years, to improve the mining efficiency of the top-coal caving faces, the cutting height has become increasingly larger. Moreover, the mining of thick coal seams generates violent disturbances such that the main roof undergoes significant rotational deformation, placing a high abutment pressure on the coal mass in front of the working face. As a result, the rib spalling of the coal wall in the working face becomes more severe. Rib spalling of the coal wall increases the tip-to-face distance, readily resulting in roof caving and falls, which not only affect the production efficiency of the working face but also tend to cause accidents with equipment and personnel [3–5]. A schematic diagrams of accidents caused by rib spalling and roof caving in the working face are shown in Figure 1. Rib spalling has become an important factor that restricts the

popularization of fully mechanized caving mining for large mining heights [6–9]. Therefore, the analysis of the factors influencing the deformation and failure of a coal wall in a fully mechanized caving face with a large mining height has important practical engineering significance for controlling the stability of coal walls and improving the production safety in the Cuncaota II Coal Mine.

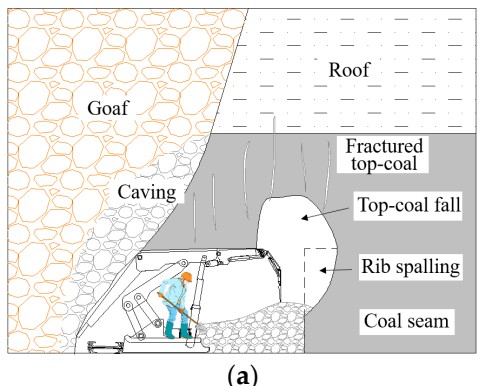 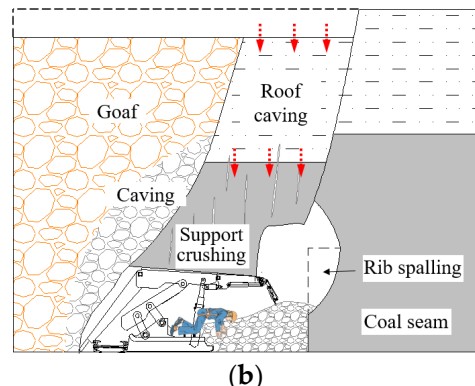

(**a**)  (**b**)

**Figure 1.** Accidents caused by rib spalling of the coal wall. (**a**) Top-coal falling to the working face. (**b**) Roof caving that induces support crushing.

At present, many scholars have thoroughly investigated the factors affecting and techniques for controlling the rib spalling of a coal wall in the working face. In terms of the trends and mechanical mechanisms of rib spalling of coal walls, Liu et al. [10,11] established a mechanical model for assessing the stability of a wedge-shaped sliding mass on a coal wall in a fully mechanized caving face with a large mining height. They studied the relationship between the stability factor of the sliding mass and the mechanical parameters of the coal mass. Yang et al. [12] analyzed the development and evolution of mining-induced fractures and revealed the mechanical process of the failure of the coal wall based on the theory of the sliding line to reveal the mechanical mechanisms of rib spalling of coal walls. Li et al. [13,14] established a mechanical model of sliding and rib spalling of coal walls based on Bishop's approach and took the safety factor of the sliding plane as an index to judge the stability of a coal mass. Wu et al. [15] theoretically analyzed the failure modes and forms of rib spalling of coal walls with a large dip angle and discussed the disaster-inducing mechanisms of the surrounding rock by the rib spalling of the coal wall by combining the spatial structure and movement characteristics of the working face. Pang and Wang [16] evaluated the stress path effects of the rib spalling of the coal wall and built a fracturing-sliding mechanical model for rib spalling of coal walls in hard, thick coal seams to determine the relationship between the failure depth and width of the coal wall and the mechanical parameters of the coal mass. Sinha and Walton [17] established a continuum model based on the progressive S-shaped yield criterion to explore the displacement and stress distribution laws in the coal wall and to predict the extent and rate of the cumulative damage to the coal wall.

Regarding the theories and technologies used for preventing and controlling the rib spalling of coal walls, Guo et al. [18] established support and surrounding rock models of a working face with a large mining height under the main roof with different structures and assessed support–roof interactions. They determined the expression for the interaction between the instability caused by the rib spalling of the coal wall and the support strength, and they reduced the rib spalling of the coal wall via advanced deep-hole grouting under static pressure. Lei et al. [19] investigated the current situation and factors influencing rib spalling in a fully mechanized working face in soft and unstable coal seams and proposed a comprehensive prevention and control technique based on the stress characteristics and rib spalling mechanisms of coal walls. Finally, the degree of rib spalling of a coal wall was reduced by optimizing the Malisan grouting parameters. By studying the factors influencing the failure of a coal wall with a fully mechanized face in ultra-thick coal seams, Wang et al. [20] identified the relationship between the strength of a coal seam and the thickness of the top

coal for different cutting heights. By combining their analysis with engineering examples, they gave the optimal cutting height for maintaining the stability of a coal wall. Zhang et al. [21] studied the working principles and stress characteristics of a sidewall-protecting structure of a hydraulic support and evaluated the delaying effects of sidewall-protecting plates on the timing of rib spalling of a coal wall. By analyzing the influences of factors pertaining to the coal mass on the stability of the coal wall, Ghadimi et al. [22,23] determined that the rib spalling of a coal wall can be prevented and controlled using methods such as grouting reinforcement of the coal wall. Mohamed et al. [24] reported that rib bolts can resist both tension and bending loads, but they cannot contain rib spall on their own, especially in weak coal seams; in such cases, external control devices are required (such as chain link, steel, and plastic) to prevent wall instability.

In previous studies, the coal wall has mostly been taken as an independent research object to explore its properties; however, the instability caused by the rib spalling of a coal wall in the top-coal caving face is the result of the comprehensive effects of multiple factors, so it is necessary to consider the influences of these multiple factors, such as mining parameters, roof (top coal), and support on the stability of a coal wall. A load-bearing mechanical model of a roof–coal wall–support system was established based on the moment–balance relationship. Applying the theory of elastic mechanics, the expressions of the deformation and stress distribution in the coal wall were solved. The horizontal displacement of a coal wall was analyzed by taking a top-coal caving face of the Cuncaota II Coal Mine in the Shendong mining area as an example. Furthermore, using an orthogonal test design and considering different levels of each factor, the horizontal displacement of a coal wall and the volume of a plastic zone were obtained through numerical simulation. On this basis, the influences of factors affecting the failure of coal walls and their rank-order according to sensitivity were obtained. Finally, by combining this with the engineering conditions prevailing in the Cuncaota II Coal Mine, the key parameters for controlling and measures for preventing the rib spalling of the coal wall were determined.

## 2. Load-Bearing Mechanical Model

### 2.1. Project Profile

The 31,204 working face in the Cuncaota II Coal Mine, located in the second panel of the #31 coal seam, is the primary working face of the mine. The working face, with a strike length of 2642 m and a dip length of 220 m, has an average burial depth of 320 m; and the dip angle of the coal seam is between 1° and 6°, making it a near-horizontal coal seam. The front section of the working face was the combination of the #31 coal seam and #42 coal seam, which had an average coal thickness of 6.87 m and a mining length of 1251 m. The fully mechanized top-coal caving process was adopted. In the rear section, the #31 coal seam, with an average coal thickness of 3.45 m, was mined for 1391 m using full-seam mining. The daily footage of the working face was 7.8 m. The general arrangement is shown in Figure 2.

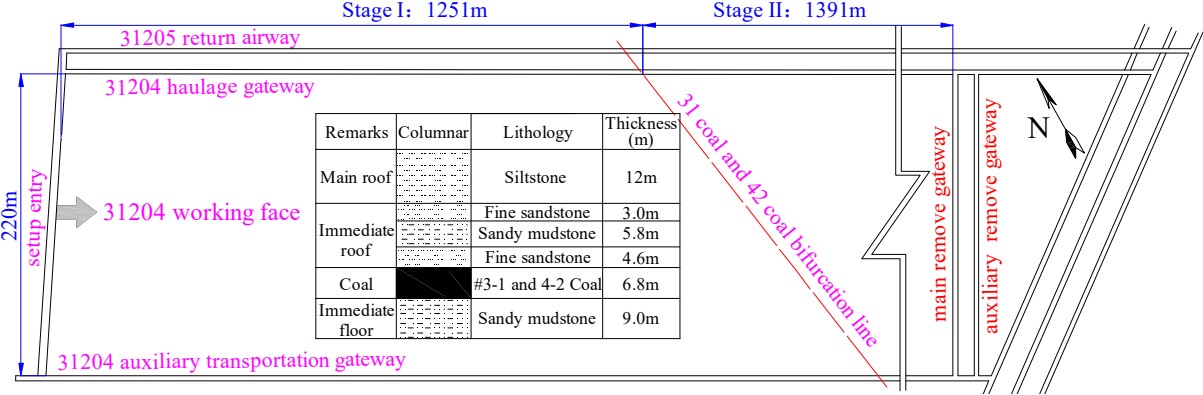

**Figure 2.** General arrangement and stratigraphy.

### 2.2. Mechanical Model

Without considering the hinge relationship of rock blocks, a mechanical model of stress on the coal wall in the fully mechanized working face was built by taking the critical state of breaking of the main roof (Figure 3) [2]. In Figure 3, point $O$ represents the point of action of the moment; $F$, $H$, $R$, $Z$, and $M$ denote the floor thickness, coal cutting height, height of top coal, thickness of the immediate roof, and thickness of the main roof, respectively; $\alpha$ and $\beta$ denote the angle of rotation of strata of the main roof and caving angle of strata, respectively; $D_1$ and $D_2$ denote the distance from the coal wall to the breaking line of the main roof and the length of top beam of the top-coal caving support, respectively; $P_1$ and $P_2$ indicate the support forces provided by coal wall and the support to the top coal, respectively.

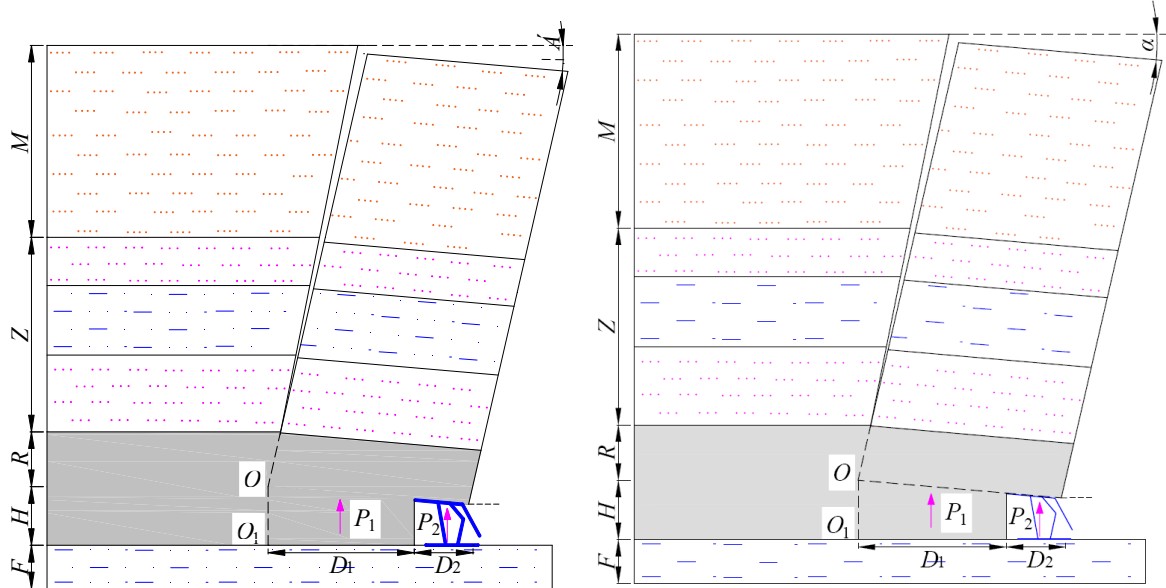

**Figure 3.** Mechanical model.

Taking the moments about $O$:

$$M_R + M_Z + M_M = M_{P_1} + M_{P_2} \tag{1}$$

where $M_R$, $M_Z$, $M_M$, $M_{P1}$, and $M_{P2}$ represent the moments of top coal, immediate roof, main roof, coal wall, and support about $O$, respectively. From Figure 3:

$$M_{P_1} = P_1 D_1 / 2 \tag{2}$$

$$M_{P_2} = P_2(D_1 + D_2 / 2) \tag{3}$$

In accordance with Figure 4,

$$M_R = \frac{l_R^2 R \rho_R g \cos \alpha}{2} \tag{4}$$

Similarly,

$$M_Z = \sum_{i=1}^{n} \frac{l_R^2 Z_i \rho_{Zi} g \cos \alpha}{2} \tag{5}$$

$$M_M = \frac{l_R^2 M \rho_M g \cos \alpha}{2} \tag{6}$$

where

$$l_R = (D_1 + D_2) / \cos \alpha$$

where $\rho_R$ and $\rho_M$ represent the densities of top coal and main roof, respectively; $n$ and $g$ represent the number of layers of the immediate roof and the acceleration due to gravity, respectively; $Z_i$ and $\rho_{zi}$ denote the thickness and density of the *i*th layer of the immediate roof, respectively.

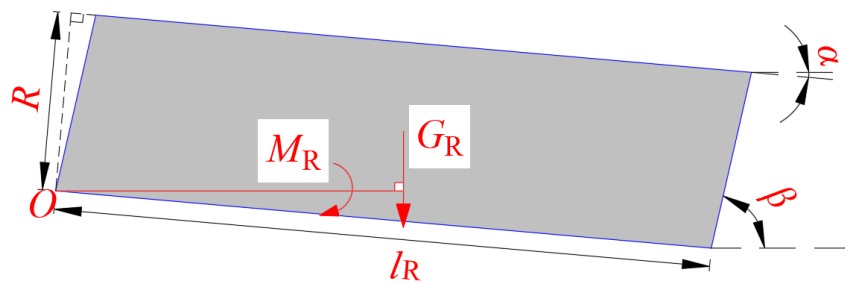

**Figure 4.** Moment analysis of top coal on point *O*.

The coal mass in a part of area in front of the coal wall has yielded, making this a problem of plastic mechanics. By reducing the elastic modulus of coal mass, it is transformed into a problem of elastic mechanics for an approximate solution.

The stiffness of the main roof and the immediate roof of the coal seam is greater than that of top coal, so it can be considered that the upper boundary of the coal wall is a given deformation boundary, while the lower and left boundaries can be regarded as fixed boundaries. *q* denotes the force of the sidewall-protecting plates of the support acting on the coal wall. The mechanical calculation model of the coal wall was established (Figure 5). The size of the coal wall in the dip direction of the working face is much larger than that along *x* and *y* directions, so this problem should be a plane-strain problem.

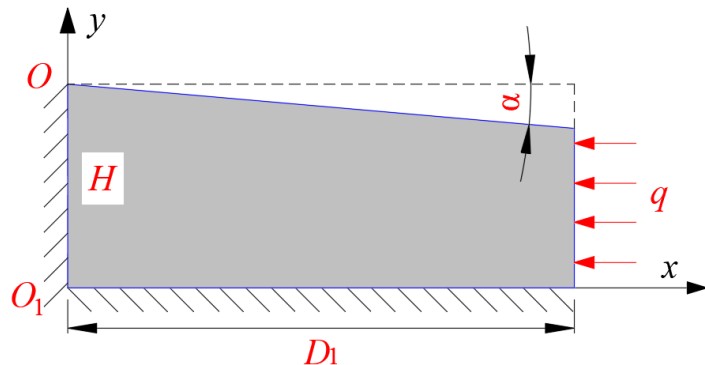

**Figure 5.** Mechanical calculation model for a coal wall.

In the plane-strain problem, the deformation potential energy $V_\varepsilon$, represented by the displacement component, can be expressed as follows:

$$V_\varepsilon = \frac{E}{2(1+\mu)} \times \int\int \left[ \frac{\mu}{1-2\mu} \left( \frac{\partial u}{\partial x} + \frac{\partial v}{\partial y} \right)^2 + \left( \frac{\partial u}{\partial x} \right)^2 + \left( \frac{\partial v}{\partial y} \right)^2 + \frac{1}{2} \left( \frac{\partial v}{\partial x} + \frac{\partial u}{\partial y} \right)^2 \right] \mathrm{d}x\mathrm{d}y \qquad (7)$$

where $E$ and $\mu$ denote the elastic modulus and Poisson's ratio of the coal mass, respectively.

If the components of elastic displacement $u$ and $v$ are subjected to small changes $\delta u$ and $\delta v$, respectively, allowed by the displacement boundary conditions, the variational formula of displacement can be obtained as follows:

$$\delta V_\varepsilon = \int\int (f_x \delta u + f_y \delta v)\mathrm{d}x\mathrm{d}y + \int (\overline{f}_x \delta u + \overline{f}_y \delta v)\mathrm{d}s \qquad (8)$$

where $f_x$ and $f_y$ indicate the components of physical force, and $\overline{f}_x$ and $\overline{f}_y$ represent the components of the surface force.

It is assumed that displacement components are expressed as follows:

$$\left.\begin{array}{l} u = u_0 + \sum\limits_{m} A_m u_m \\ v = v_0 + \sum\limits_{m} B_m v_m \end{array}\right\} \tag{9}$$

where $A_m$ and $B_m$ indicate the $2m$ independent coefficients; $u_0$, $v_0$, $u_m$, and $v_m$ denote the set coordinate functions. On the displacement boundary, $u_0$ and $v_0$ are equal to the given boundary values $\overline{u}$ and $\overline{v}$, respectively; and $u_m$ and $v_m$ are zero.

By differentiating Equation (9), the variation in the deformation potential energy is obtained and substituted into Equation (8), and the following equation can be obtained:

$$\left.\begin{array}{l} \frac{\delta V_\varepsilon}{\partial A_m} = \iint f_x u_m \mathrm{d}x\mathrm{d}y + \int \overline{f}_x u_m \mathrm{d}s \\ \frac{\delta V_\varepsilon}{\partial B_m} = \iint f_y v_m \mathrm{d}x\mathrm{d}y + \int \overline{f}_y v_m \mathrm{d}s \end{array}\right\} \tag{10}$$

Based on the mechanical model of the coal wall, the boundary conditions were determined as follows:

(1)  The components of physical force are ignored, that is, $f_x = 0$ and $f_y = 0$.
(2)  The boundary conditions of surface force are such that when $x = D_1$, $\overline{f}_x = -q$, and $\overline{f}_y = 0$.
(3)  The displacement boundary conditions are such that when $x = 0$, $\mu = v = 0$; when $y = 0$, $u = v = 0$; when $y = H$, $v = -x \tan \alpha$.

The method of displacement variation was used for calculation. It is supposed that the displacement components meeting the boundary conditions are expressed as follows:

$$\left.\begin{array}{l} u = A_1(xy + xy(H - y)) \\ v = -xy\frac{\tan\alpha}{H} + B_1 xy(\frac{y}{H} - 1) \end{array}\right\} \tag{11}$$

where $A_1$ and $B_1$ represent the undetermined coefficients.

By substituting the boundary conditions and Equation (11) into Equation (10), the following formula is obtained:

$$\left.\begin{array}{l} \frac{\delta V_\varepsilon}{\partial A_1} = -\frac{1}{6}qD_1 H^2(3 + H) \\ \frac{\delta V_\varepsilon}{\partial B_1} = 0 \end{array}\right\} \tag{12}$$

By substituting Equation (11) into Equation (7) for integration, the derivatives of $A_1$ and $B_1$ are taken as follows:

$$\left.\begin{array}{l} \frac{\delta V_\varepsilon}{\delta A_1} = \frac{EHD_1}{360(\mu+1)(2\mu-1)}[15D_1(B_1 H(-1 + 4\mu) + (3 - H + 4H\mu)\tan\alpha) + \\ \quad 4A_1(3H^4(-1 + \mu) + 15H^3(-1 + \mu) + 15D_1^2(-1 + 2\mu) + \\ \quad 5H^2(6(-1 + \mu) + D_1^2(-1 + 2\mu)))] \\ \frac{\delta V_\varepsilon}{\delta B_1} = -\frac{EHD_1}{360(\mu+1)(2\mu-1)}[15H((1 - 2\mu)\tan\alpha) + A_1 D_1(-1 + 4\mu)) + \\ \quad 2B_1(20D_1^2(1 - \mu) + 3H^2(1 - 2\mu))] \end{array}\right\} \tag{13}$$

By combining Equation (12) with Equation (13), the expressions for $A_1$ and $B_1$ can be obtained. According to elastic mechanics, the distributions of horizontal stress and vertical stress in front of the working face can be expressed as follows:

$$\left.\begin{array}{l} \sigma_x = \frac{E}{1+\mu}\left(\frac{\mu}{1-2\mu}\left(\frac{\partial u}{\partial x} + \frac{\partial v}{\partial y}\right) + \frac{\partial u}{\partial x}\right) \\ \sigma_y = \frac{E}{1+\mu}\left(\frac{\mu}{1-2\mu}\left(\frac{\partial u}{\partial x} + \frac{\partial v}{\partial y}\right) + \frac{\partial v}{\partial y}\right) \end{array}\right\} \tag{14}$$

By substituting Equation (11) into Equation (14), the stress in the coal wall can be obtained. The moment of the coal wall at point $O$ can be expressed as follows:

$$M_{P_1} = \overline{\sigma_y}D_1 = (\sigma_{y|x=0,y=H} + \sigma_{y|x=D_1,y=H})D_1/2 \tag{15}$$

By simultaneously solving Equations (1) and (15), $\alpha$ can be obtained. By substituting $A_1$, $B_1$, and $\alpha$ into Equations (11) and (14), the expressions of displacement and stress distribution in the coal wall can be obtained.

### 2.3. Analysis of Factors Influencing Failure of Coal Wall

By analyzing the expressions of the displacement and stress distribution in coal walls, the factors inducing the failure of the coal wall can be classified into three categories.

(1)  Factors influencing occurrence, which refer to the natural occurrence conditions of coal seams, mainly including the thicknesses and densities of the main roof, immediate roof in each layer, and coal seams.

(2)  Internal influencing factors, which are the physical and mechanical properties of the coal mass, such as the elastic modulus and Poisson's ratio. In practice, a coal mass is an elastoplastic body, so its cohesion and angle of internal friction should also be considered.

(3)  External influencing factors, that is, external disturbances subjected by the coal mass, such as the cutting height (or height of top coal), breaking position of the main roof, length of the top beam of the support, support strength, and sidewall-protecting force of the support.

For large-scale mines in the west, such as the Cuncaota II Coal Mine, the dip angle of the strata is nearly horizontal, the thickness of coal seams does not vary to any significant extent, and its geological structure is simple. Therefore, the factors influencing the occurrence, such as basic geological conditions, are largely unaffected by the mining process. The internal influencing factors can be mitigated by advanced grouting; however, the working face in the modern high-yield and high-efficiency mines advances quickly and the process, including advanced grouting, takes a long time and has a small range of action. Such a process is not suitable for large-scale application to the working face but can be used as an auxiliary measure under severe rib spalling; therefore, human control slightly changes the internal influencing factors. In conclusion, to prevent the rib spalling of the coal wall, the external influencing factors are the only factors that can be controlled by human action.

According to the strata occurrence and prevailing mining conditions at the 31,204 working face, $\rho_R$ = 1280 kg/m³, $\rho_Z$ = 2440 kg/m³, $\rho_M$ = 2675 kg/m³, $Z$ = 13.4 m, $M$ = 12 m, $E$ = 18 MPa, and $\mu$ = 0.3. By changing the parameters of the external influence factors, the horizontal displacement of the coal wall was plotted (Figure 6). The figure shows the horizontal displacement of the coal wall under the effects of different external influencing parameters.

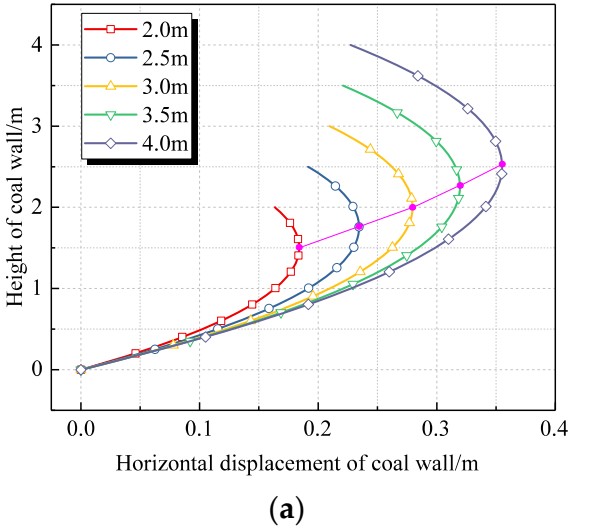

**(a)**

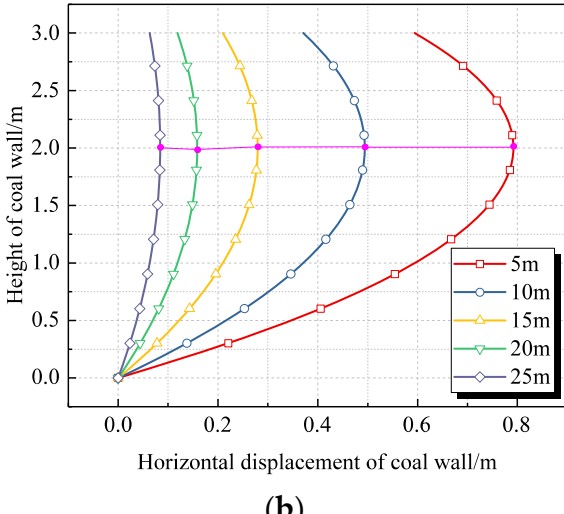

**(b)**

**Figure 6.** *Cont.*

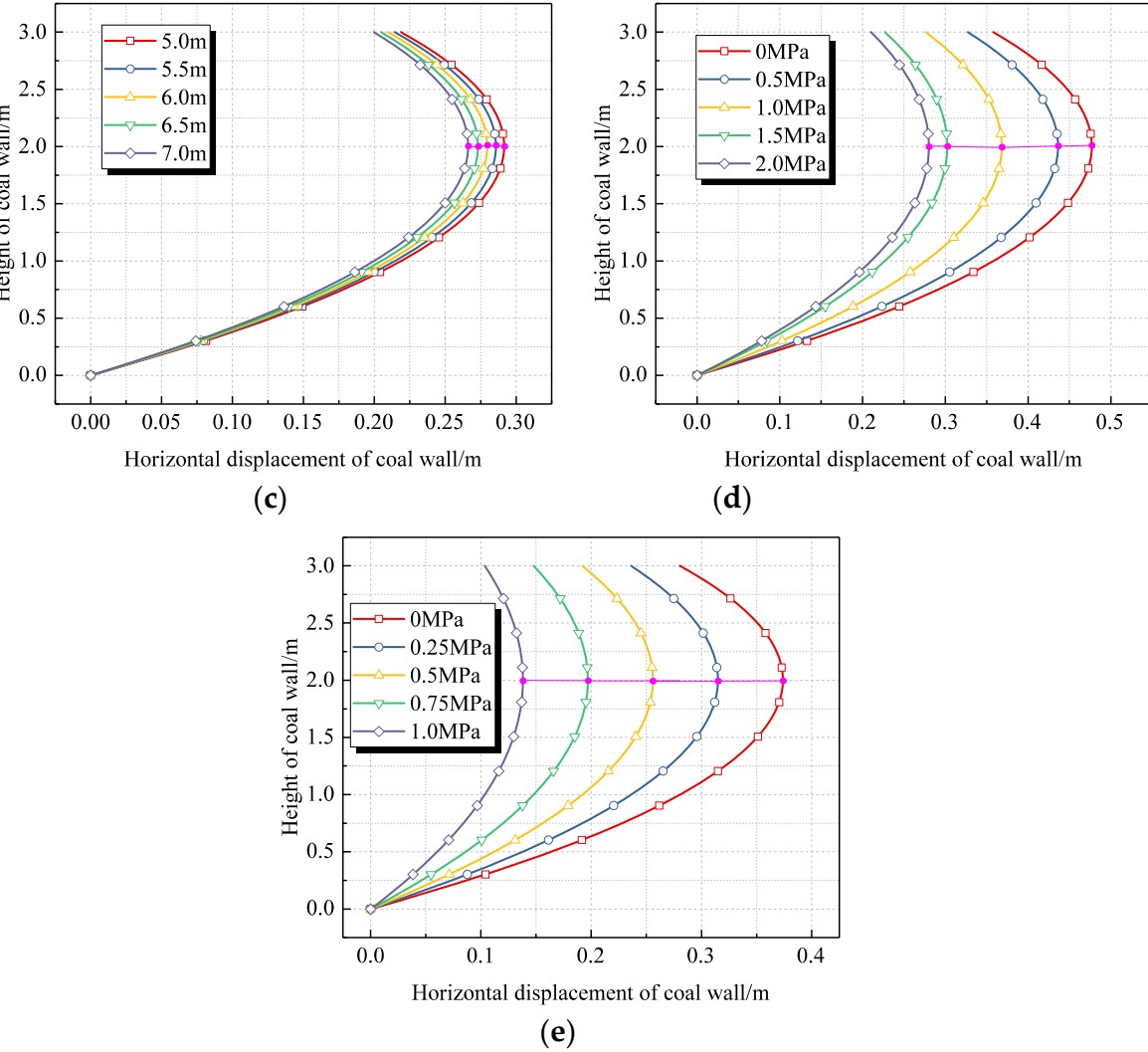

**Figure 6.** Horizontal displacement of the coal wall under effects of different external influencing factors. (**a**) Cutting height of coal seams. (**b**) Breaking position of the main roof. (**c**) Length of top beam of the support. (**d**) Support strength. (**e**) Sidewall-protecting force.

As shown in Figure 6,

(1) Under the influences of the same factor, the characteristics of the distribution of the horizontal displacement of the coal wall along the height of the coal wall are similar; namely, the horizontal displacement is small in the upper and lower parts and large in the middle. The maximum horizontal displacement appears in the upper middle in terms of the height of the coal wall.

(2) When the cutting height is 3 m, the maximum horizontal displacement of the coal wall appears at a height of about 2 m. When the other factors remain unchanged, with the increase in the cutting height of the coal seam, the stability of the coal wall is decreased, so that the horizontal displacement of the coal wall rises. When the cutting height of the coal seam increases from 2 m to 4 m, the maximum horizontal displacement of the coal wall rises from 0.18 m to 0.35 m (Figure 6a).

(3) The closer the breaking position of the main roof to the coal wall, the greater the pressure on the coal wall, leading to increased horizontal displacement. As the working face advances, the coal wall constantly approaches the breaking line of the main roof, and the horizontal displacement significantly increases, thus increasing the risk of the rib spalling of the coal wall. As shown in Figure 6b, when the distance from the breaking position of the main roof to the coal wall decreases from 25 m to

5 m, the maximum horizontal displacement of the coal wall increases from 0.08 m to 0.8 m. Moreover, with the decrease in that distance, the displacement significantly increases. When the main roof is weighted, the rib spalling of the coal wall is rapidly aggravated, which is consistent with experience.

(4) As the length of top beam of the support increases, the horizontal displacement of the coal wall decreases, but the change is insignificant. As shown in Figure 6c, when the length of top beam rises from 5 m to 7 m, the maximum horizontal displacement of the coal wall decreases only by 30 mm from 0.29 m to 0.26 m. This is because, on the one hand, as the top beam of the support increases in length, the roof-control length and instability of the coal wall increase; on the other hand, the increase in the length of top beam enhances the support force, which is conducive to stabilizing the coal wall. These counteract each other, so that the change in the length of the top beam of the support has a little influence on the failure of the coal wall.

(5) With the increase in the support strength, the pressure of the overlying strata undertaken by the coal wall decreases, and the horizontal displacement of the coal wall reduces. As shown in Figure 6d, when the support strength is increased from 0 to 1.5 MPa, the maximum horizontal displacement decreases from 0.48 to 0.30 m. The horizontal displacement significantly decreases with increasing support strength. As the support strength is increased from 1.5 to 2.0 MPa, the maximum horizontal displacement decreases from 0.30 to 0.28 m. That is, with the further increase in the support strength, the horizontal displacement slightly reduces, indicating that 1.5 MPa is the critical support strength. Below this strength, the horizontal displacement of the coal wall is sensitive to changes in the support strength; above this strength, it is insensitive to changes in the support strength.

(6) The sidewall-protecting force of the support directly acts on the coal wall and exerts an important influence on the horizontal displacement of the coal wall. With the increase in the sidewall-protecting force afforded by the support, the horizontal displacement of the coal wall decreases. As shown in Figure 6e, the maximum horizontal displacement of and the maximum displacement variation in the coal wall occur at a height of 2.0 m, which can be used as the boundary point of the influence of the sidewall-protecting force on the horizontal deformation of the coal wall. Along the height of the coal wall, the further from that point, the less sensitive the horizontal deformation of the coal wall to the sidewall-protecting force. When the support strengths are 0 and 2 MPa, at a height of 2.0 m, the horizontal displacements are 0.37 and 0.14 m (a difference of 0.23 m), respectively; at a height of 1.0 m, the horizontal displacements are 0.28 and 0.10 m (a difference of 0.18 m), respectively. This suggests that the control effects of the sidewall-protecting force on the coal wall are diminished.

## 3. Research Methods

### 3.1. Orthogonal Test Schemes

The failure of a coal wall is the result of the interactions of multiple influencing factors. Theoretical analysis fails to fully reflect the effects and influences of various factors on the failure of a coal wall, so the numerical simulation method was used for analysis.

If multiple levels of each influence factor are analyzed one-by-one, the computational burden would be onerous, and the sensitivity of the failure of the coal wall to various influence factors would not be comparable. Orthogonal testing, as a powerful method of multifactor analysis, can be used to select some representative testing points that are evenly dispersed, homogeneous, and comparable; moreover, it can reveal the optimal collocation of levels of the factors with only a few tests or be used to infer the optimal collocation from the test results through calculation [25,26].

In accordance with engineering practice and theoretical analysis, the length of top beam of the support changes little, having little effect on the horizontal displacement of the coal wall. Therefore, four factors, namely, the cutting height of the coal seams, distance

from the breaking position of the main roof to the coal wall, support strength, and sidewall-protecting force, were only considered in the orthogonal test; each factor was set to one of four levels. It is noteworthy that the sum of the cutting height and thickness of the top coal was equal to the total coal thickness; that is, with the increase in the cutting height, the thickness of top coal correspondingly decreases. By using SPSS software (Version 26), an $L_{16}(4)^5$ orthogonal table was selected, and only 16 tests were needed (Table 1).

**Table 1.** Levels of factors in the orthogonal test.

| Level | Factor | | | |
|---|---|---|---|---|
| | Cutting Height/m | Breaking Position of Main Roof/m | Support Strength/MPa | Sidewall Protecting Force/MPa |
| 1 | 2.5 | 5 | 0 | 0 |
| 2 | 3.0 | 10 | 0.5 | 0.25 |
| 3 | 3.5 | 15 | 1.0 | 0.5 |
| 4 | 4.0 | 20 | 1.5 | 0.75 |

### 3.2. Model Establishment

Based on the prevailing geological conditions around the 31,204 working face in the Cuncaota II Coal Mine, a model of the fully mechanized caving face was established using Flac$^{3D}$ numerical simulation software to conduct sensitivity analysis on the failure of the coal wall to various influencing factors. The established model measured 100 m × 280 m × 50 m (length × width × height). The self-weight of the overlying strata was replaced with a uniformly distributed load applied to the top; both the sides and the bottom of the model were limited by displacement-controlled boundary conditions. The support strength and sidewall-protecting force of the support were represented by an equivalent applied load. The main roof was broken by inserting a vertical weak plane in the main roof at a position a certain distance in front of the working face. In the initial equilibrium state of the model, the parameters of the main roof were assigned to the weak plane. When the coal seam was excavated to the designated position, the weak plane was reassigned to reduce the relevant parameters including cohesion, thus simulating the breaking of the main roof. The model is displayed in Figure 7. According to the test schemes (Table 2), the simulation was conducted. The horizontal displacement of the coal wall and the volume of the plastic zone in the coal wall in unit length of the dip direction were taken as the indices for evaluating the rib spalling of the coal wall [27,28]. The specific schemes and results of the orthogonal tests are summarized in Table 2.

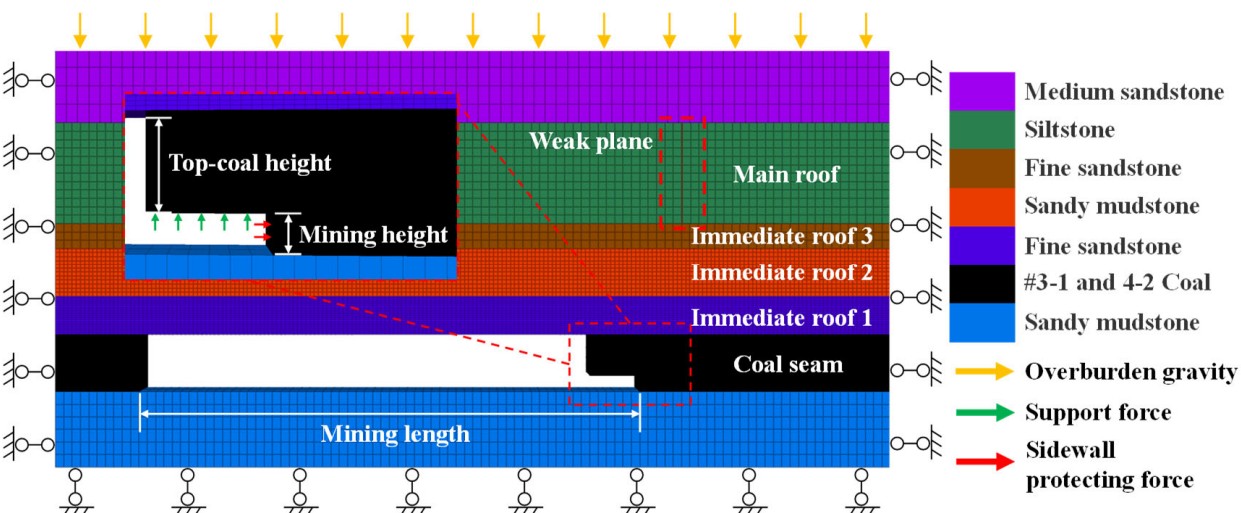

**Figure 7.** Simulation model.

**Table 2.** Schemes and results of orthogonal tests with multiple factors.

| Test Serial Number | Cutting Height/m | Breaking Position of Main Roof/m | Support Strength/MPa | Sidewall Protecting Force/MPa | Horizontal Displacement/mm | Volume of Plastic Zone/m³ |
|---|---|---|---|---|---|---|
| i-1 | 2.5 | 5 | 0 | 0 | 423.61 | 12.25 |
| i-2 | 2.5 | 10 | 1 | 0.25 | 275.15 | 7 |
| i-3 | 2.5 | 15 | 1.5 | 0.5 | 201.73 | 5.13 |
| i-4 | 2.5 | 20 | 0.5 | 0.75 | 175.97 | 5.06 |
| i-5 | 3.0 | 5 | 0.5 | 0.25 | 508.27 | 16.17 |
| i-6 | 3.0 | 10 | 1.5 | 0 | 333.38 | 10.69 |
| i-7 | 3.0 | 15 | 1 | 0.75 | 251.41 | 8.79 |
| i-8 | 3.0 | 20 | 0 | 0.5 | 266.78 | 9.44 |
| i-9 | 3.5 | 5 | 1 | 0.5 | 548.12 | 19.06 |
| i-10 | 3.5 | 10 | 0 | 0.75 | 379.37 | 15.63 |
| i-11 | 3.5 | 15 | 0.5 | 0 | 410.02 | 13.75 |
| i-12 | 3.5 | 20 | 1.5 | 0.25 | 281.92 | 11.25 |
| i-13 | 4.0 | 5 | 1.5 | 0.75 | 607.17 | 20.75 |
| i-14 | 4.0 | 10 | 0.5 | 0.5 | 425.01 | 18.81 |
| i-15 | 4.0 | 15 | 0 | 0.25 | 405.5 | 17.70 |
| i-16 | 4.0 | 20 | 1 | 0 | 407.44 | 15.83 |

## 4. Results

### 4.1. Range Analysis

The range of a parameter can be used to evaluate the significance of factors, and the amplitude of change therein represents the degree of influence of the changes in the levels of a factor on the dependent variables. A larger range indicates greater influences of changes in the levels of the factor on the dependent variables. Range analysis is qualitative [29,30]; in this study, the horizontal displacement of the coal wall and the volume of the plastic zone in the coal wall in the unit length of the dip direction were taken as dependent variables. The range analysis results are listed in Table 3.

**Table 3.** Range analysis results.

| Index | Factor | $\overline{K}_{1j}$ | $\overline{K}_{2j}$ | $\overline{K}_{3j}$ | $\overline{K}_{4j}$ | $\overline{R}_N$ | Rank of Influence Level |
|---|---|---|---|---|---|---|---|
| Horizontal displacement | Cutting height | 269.12 | 339.96 | 404.86 | 461.28 | 192.17 | • Breaking position of main roof |
| | Breaking position of main roof | 521.79 | 353.23 | 317.17 | 283.03 | 238.77 | • Cutting height |
| | Support strength | 368.82 | 379.82 | 370.53 | 356.05 | 23.77 | • Sidewall-protecting force |
| | Sidewall protecting force | 393.61 | 367.71 | 360.41 | 353.48 | 40.13 | • Support strength |
| Volume of plastic zone | Cutting height | 7.36 | 11.27 | 14.92 | 18.27 | 10.91 | • Cutting height |
| | Breaking position of main roof | 17.06 | 13.03 | 11.34 | 10.40 | 6.66 | • Breaking position of main roof |
| | Support strength | 13.76 | 13.45 | 12.67 | 11.96 | 1.8 | • Support strength |
| | Sidewall protecting force | 13.13 | 13.03 | 13.11 | 12.56 | 0.57 | • Sidewall-protecting force |

Based on the data in Table 3, by taking the levels of each influencing factor as the horizontal coordinate and the averages of range between the horizontal displacement and the volume of the plastic zone as the vertical coordinate, curves were plotted (Figure 8).

As demonstrated in Table 3 and Figure 8:

(1) The rank-order of the factors in terms of their influences on the horizontal displacement of the coal wall and the volume of the plastic zone differs; however, the ranges of the cutting height and the breaking position of the main roof are greater than those of the other factors, indicating that they have a greater (and more sensitive) influence on the horizontal displacement of the coal wall and the volume of the plastic zone. The support strength and sidewall-protecting force of the support are less influential. The factors were ranked in the order of the breaking position of the main roof, cutting height, sidewall-protecting force, and support strength in terms of influences on the horizontal displacement of the coal wall. The breaking position of the main roof plays a dominant role in affecting the horizontal displacement, followed by the cutting

height. The factors were ranked in descending order as the cutting height, breaking position of the main roof, support strength, and sidewall-protecting force regarding their influences on the volume of the plastic zone. The cutting height plays a dominant role in affecting the volume of the plastic zone, followed by the breaking position of the main roof.

(2) By affecting the thickness of the top coal and the ranges of influence of the abutment pressure, the cutting height changes the horizontal displacement of the coal wall and the volume of the plastic zone. With the increase in the cutting height, the two dependent variables are positively correlated because a larger cutting height decreases the thickness of top coal and the cushioning effects of the weak top coal, and the mine pressure directly acts on the coal wall, thus aggravating the instability of the coal wall. Furthermore, with the increase in the cutting height, the range of influence of the abutment pressure in front of the coal wall extends, so that fractures develop in the coal mass, and the range of plastic failure zone grows; therefore, the cutting height plays a leading role in affecting the volume of the plastic zone.

(3) By affecting the stress in the coal wall, the breaking position of the main roof changes the horizontal displacement of the coal wall and the volume of the plastic zone. When the distance from the breaking position of the main roof to the coal wall exceeds 10 m, the two dependent variables gradually increase as the breaking position approaches the coal wall. When the distance from the breaking position to the coal wall is less than 10 m, the two dependent variables rapidly increase. By combining this with Equation (1), to ensure the stability of the structure, the force ($P_1$) provided by the coal wall needs to increase when the breaking position of the main roof is closer to the coal wall ($D_1$ decreases). In this case, a larger pressure applied on the coal wall leads to the extrusion of the coal wall and the constant accumulation of horizontal displacement. On this basis, it was determined that there is a certain critical value for the breaking position of the main roof. When the distance from the position to the coal wall exceeds the critical value, the stress applied to the coal wall is small, and the horizontal displacement is small, so the coal wall can remain stable. When the distance is less than the critical value, the horizontal displacement of the coal wall significantly increases, so the coal wall tends to be unstable. Therefore, the breaking position of the main roof plays a leading role in influencing the horizontal displacement.

(4) The support strength and sidewall-protecting force have limited effects on the horizontal displacement of the coal wall and the volume of the plastic zone. With increasing support strength and sidewall-protecting force, the two dependent variables tend to decrease (albeit marginally). In engineering practice, the support strength and sidewall-protecting force are much less than the mine pressure, which slightly affects the dependent variables; however, because the hydraulic support directly act on the coal mining space (the sidewall-protecting force especially directly acts on the coal wall), their influences on the horizontal displacement are somewhat stronger.

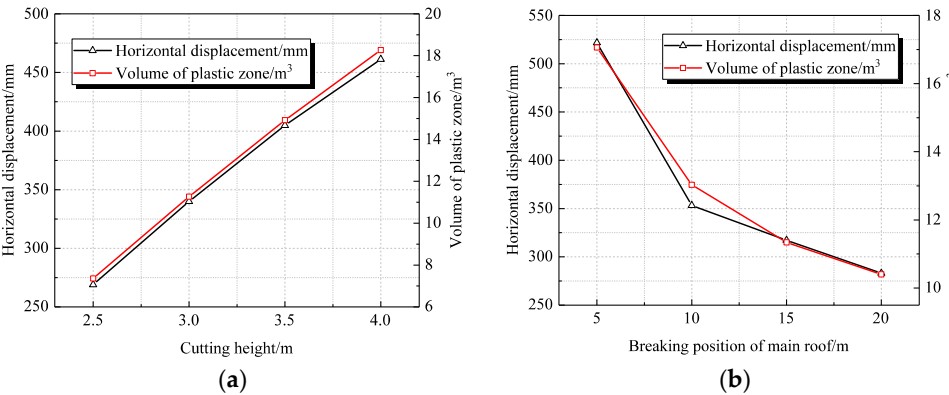

**Figure 8.** *Cont.*

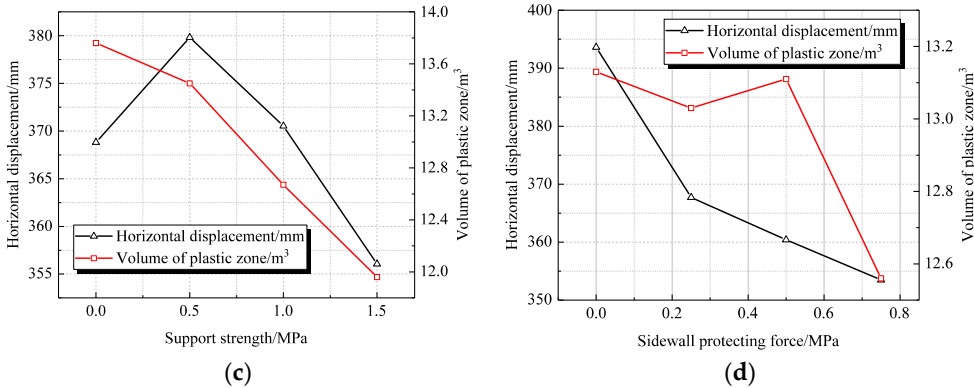

**Figure 8.** The change curves of the horizontal displacement and the volume of the plastic zone for various factors. (**a**) Cutting height. (**b**) Breaking position of main roof. (**c**) Support strength. (**d**) Sidewall-protecting force.

### 4.2. Variance Analysis

Analysis of variance is a quantitative technique that can determine the degrees of influence and confidence of each factor on the performance results [29,31]. By comparing the average sum of the square of the deviations between factors and errors, $F$ values are obtained, which reflect the extent of the influence of changes in levels of factors on the test indices [32,33]. Based on the given significance level $\alpha$ (confidence), the critical value $F_\alpha$ $(f_f, f_e)$ is sought from the $F$-distribution table according to the degrees of freedom. The determination of the significance of influences is demonstrated in Table 4 [34]. The $F$ value is given by:

$$F = \frac{S_f / f_f}{S_e / f_e} \tag{16}$$

where, $S_f$ and $f_f$ represent the sum of the squares of the deviations and the number of degrees of freedom of each factor, respectively; and $S_e$ and $f_e$ represent the sum of the squares of the deviations and number of degree of freedom of errors in the test, respectively.

**Table 4.** Determination of level of significance.

| Serial Number | Range of $F$ | Sensitivity Statement | Degree |
|:---:|:---:|:---:|:---:|
| 1 | $F > F_{0·001}(f_f, f_e)$ | Highly significant effects | *** |
| 2 | $F_{0·001}(f_f, f_e) > F > F_{0·01}(f_f, f_e)$ | Significant effects | ** |
| 3 | $F_{0·01}(f_f, f_e) > F > F_{0·05}(f_f, f_e)$ | General effects | * |
| 4 | $F < F_{0·05}(f_f, f_e)$ | Insignificant effects | |

Notes: $F_{0·001}(f_f, f_e) = 141.1$, $F_{0·01}(f_f, f_e) = 29.46$, and $F_{0·05}(f_f, f_e) = 9.28$.

The analysis of variance was conducted on the orthogonal test results with SPSS statistical analysis software (Table 5).

**Table 5.** Results of analysis of variance.

| Index | Source Variance | Sum of the Squared Deviation | Degrees of Freedom | $F$ | Significance |
|:---:|:---:|:---:|:---:|:---:|:---:|
| Horizontal displacement | Cutting height | 82,486.15 | 3 | 20.17 | * |
| | Breaking position of main roof | 134,689.21 | 3 | 32.94 | ** |
| | Support strength | 1147.76 | 3 | 0.28 | |
| | Sidewall-protecting force | 3687.77 | 3 | 0.90 | |
| Volume of plastic zone | Cutting height | 265.15 | 3 | 400.07 | *** |
| | Breaking position of main roof | 103.96 | 3 | 156.85 | *** |
| | Support strength | 7.87 | 3 | 11.88 | * |
| | Sidewall-protecting force | 0.87 | 3 | 1.30 | |

As shown in Table 5:

(1) In terms of the horizontal displacement, the breaking position of the main roof has significant effects, while the cutting height exerts a more moderate, general influence (the other factors exert insignificant effects). According to the sensitivity, the breaking position of the main roof, cutting height, sidewall-protecting force, and support strength are ranked thus (in descending order).

(2) For the volume of the plastic zone, the cutting height and the breaking position of the main roof have highly significant influences, and the significance of the former is greater than that of the latter. Furthermore, the support strength has general effects, while the other factors exert insignificant influences. In accordance with their sensitivity, the cutting height, breaking position of the main roof, support strength, and sidewall-protecting force are ranked in descending order.

(3) Based on the results of analysis of variance and range analysis, the factors are ranked in same order according to the sensitivity, verifying that the orthogonal test results are correct.

## 5. Discussion

Based on the above research results, the cutting height and breaking position of the main roof were found to significantly affect the rib spalling of coal walls, while the support strength and sidewall-protecting force only slightly influence rib spalling. With the increase in the cutting height, the thickness of the top coal decreases, and the cushioning effects of the weak top coal are reduced. The mine pressure directly acts on the coal wall, thus aggravating the instability of the coal wall. Additionally, the larger the cutting height, the greater the range of influence of the abutment pressure in front of the coal wall, leading to the development of fractures in the coal mass and increases in the extent of the plastic failure zone. Therefore, the cutting height plays a leading role in affecting the volume of the plastic zone. As the breaking position of the main roof gradually approaches the coal wall, the pressure on the coal wall increases, resulting in the extrusion of the coal wall and the continuous accumulation of horizontal displacement, so the breaking position of the main roof plays an important role in influencing the horizontal displacement. The support strength and sidewall-protecting force have limited effects on the rib spalling of coal walls; however, the hydraulic support directly acts on the coal-mining space; and, in particular, the sidewall protecting force directly acts on the coal wall, so the influences on the horizontal displacement are moderately strong.

Based on the above analysis, considering the engineering conditions around the 31,204 working face, the key parameters for controlling and measures for preventing the rib spalling of the coal wall are proposed:

(1) The cutting height should be reasonably controlled. When considering the ratio of the cutting height to the caving height and mining speed, an appropriate cutting height should be selected. An increase in the cutting height aggravates the occurrence of mine pressure and reduces the inherent stability of the coal wall, greatly increasing the probability of the rib spalling of the coal wall. A small cutting height increases the height of coal caving and slows the advance of the working face, and an overly small cutting height does not meet the requirement for the ratio of the cutting height to the caving height. Therefore, the cutting height of the working face should be kept to below 3.5 m and should be properly reduced in the case of severe rib spalling of the coal wall.

(2) The working resistance and sidewall-protecting force of the support should be improved. The selection of appropriate support equipment and processes can reduce the degree of rib spalling of the coal wall to some extent; therefore, the support parameters should be selected as follows: the support strength and sidewall-protecting strength should be 1.5 MPa and 0.75 MPa, respectively. Furthermore, considering that the #31 coal seam is independently mined in the rear section of the working face, the

support height should not be too small, so a ZFY18000/25/39D two-column shield hydraulic support for coal caving should be selected.

(3)  The management of the coal wall should be strengthened during weighting. The breaking of the main roof significantly affects the failure of the coal wall. Therefore, coal walls are extremely prone to rib spalling and roof caving during weighting. It is necessary to standardize the operational processes and the use of equipment at the working face to ensure the requisite initial support force and working resistance. Moreover, the roof and the coal wall (after coal cutting) are quickly supported to reduce the exposure time, and the hydraulic support of roof wiping should be advanced with the appropriate pressures applied.

By implementing the aforementioned measures, the rib spalling of the coal wall in the working face can be effectively controlled, reducing the risk of rib-spalling accidents and realizing safe, high-efficiency production at the 31,204 fully mechanized caving face over its large mining height.

## 6. Conclusions

(1)  Three categories of factors influencing the failure of coal walls were determined. The factors that could be controlled by human actions are the cutting height (or height of top coal), the breaking position of the main roof, the length of top beam of the support, support strength, and the sidewall-protecting force. The changes in the horizontal displacement of the coal wall under the influences of different values of each factor were assessed.

(2)  The order of each factor in influencing the horizontal displacement of the coal wall and the volume of the plastic zone was determined through range analysis. The breaking position of the main roof plays a leading role in the horizontal displacement, followed by the cutting height. The cutting height plays a dominant role in affecting the volume of the plastic zone, followed by the breaking position of the main roof. The support strength and sidewall-protecting force have limited effects on the horizontal displacement of the coal wall and the volume of the plastic zone.

(3)  The significance of the influence of the factors on the horizontal displacement of the coal wall and the volume of the plastic zone was determined through analysis of variance. In terms of the horizontal displacement, the breaking position of the main roof has the most significant effect, while the cutting height exerts a more moderate influence. As for the volume of the plastic zone, the cutting height and breaking position of the main roof exert highly significant influences, and the significance of the cutting height is greater than that of the breaking position of the main roof (the strength of the support exerts a more moderate, general effect).

(4)  Technical measures, such as reasonably controlling the cutting height of the working face, increasing the working resistance and sidewall-protecting force of the support, and strengthening the management of the coal wall during weighting were proposed. The rib spalling of the coal wall in the 31,204 working face can be effectively controlled, allowing the safe, high-efficiency mining of the fully mechanized caving face over its large mining height.

**Author Contributions:** Conceptualization, G.M.; Methodology, G.M.; Supervision, J.Z.; Resources, C.W.; Writing—Review and Editing, G.M., C.W. and M.L.; Methodology, J.Z., N.Z. and G.M.; Formal Analysis, N.Z., C.W. and M.L.; Funding Acquisition, N.Z. and M.L. All authors have read and agreed to the published version of the manuscript.

**Funding:** The research described in this paper was supported by the Ordos Science & Technology Plan (grant number 2022EEDSKJZDZX005) and the Postgraduate Research & Practice Innovation Program of Jiangsu Province (grant number KYCX21_2372).

**Institutional Review Board Statement:** Not applicable.

**Informed Consent Statement:** Not applicable.

**Data Availability Statement:** The data used to support the findings of this study are included within the article.

**Conflicts of Interest:** The authors declare no conflict of interest.

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
