# Peer review of "Analysis of Influencing Factors and Prevention of Coal Wall Deformation and Failure of Coal Wall in Caving Face with Large Mining Height: Case Study"

_applsci, doi:10.3390/app13127173_

Round 1

Reviewer 1 Report

Based on a careful analysis, I can formulate the following remarks:

1) The aim of this article, based on the authors’ scrupulous investigations, is to propose a load-bearing mechanical model of the roof–coal wall–support system, based on the moment balance relationship. In addition, in order to assess the risk of coal wall deformation and failure, as well as to improve the prediction, and control capacity for rib spalling, the authors calculated the expressions for the deformation and stress distribution in coal wall.

2) The topic represents in my opinion a relevant approach of the proposed theme in the field, based on meticulous theoretical investigations, correlated with simulation results.

3) In comparison with other published material, the authors' contribution adds to the subject area a new approach/methodology, with several significant contributions.

They performed a scrupulous a theoretical analysis, validated by meticulous simulation results.

It is well-known fact that the coal wall in caving face with large mining height is prone to rib spalling, which leads to phased cessation of mining of working face and causes heavy losses, and endangers the safety of underground workers.

The authors, by means of their original load-bearing mechanical model, based on the moment balance relationship, analyzed the influences of key factors on the horizontal displacement of coal wall.

It is well-known fact that the failure of the coal wall is the result of interactions of multiple influence factors. Theoretical analysis fails to reflect fully the effects and influences of various factors on the failure of the coal wall, so for an efficient analysis the numerical simulation method is the most suitable.

If multiple levels of each influence factor were analyzed one-by-one, the computational burden is onerous, and the sensitivity of the failure of the coal wall to various influence factors cannot be compared.

Orthogonal testing, as a powerful method of multi-factor analysis, can select some representative testing points which are evenly dispersed, homogeneous, and comparable; moreover, it can reveal the optimal collocation of levels of the factors with only a few tests or infer the optimal collocation from the test results through calculation. Due to this fact, the authors performed an orthogonal test design in order to finalize their analysis.

On this basis, the influences of four factors: cutting height, breaking position of main roof, support strength, and sidewall protecting force of support on the horizontal displacement and volume of a plastic zone of coal wall were analyzed. Moreover, the authors ranked by sensitivity their order of importance.

Based on engineering conditions and production practice of a given mine, key parameters for controlling and measures for preventing rib spalling of the coal wall were proposed to good practical effect.

The research results have practical significance. 

4) The thick coal seams are widely distributed in China, for more than 40% of the total coal output. The fully mechanized top-coal caving technology in thick coal seams is an important technical guarantee to improve the yield per unit of the working face.

In order to improve the mining efficiency of top-coal caving faces, the cutting height becomes increasingly large. Moreover, mining of thick coal seams generates violent disturbance such that the main roof undergoes significant rotational deformation, imposing a high abutment pressure in coal mass in front of the working face. As a result, rib spalling of the coal wall in the working face becomes more severe. Rib spalling of the coal wall increases the tip-to-face distance, readily resulting in roof caving and falls, which not only affects the production efficiency of the working face, but also tends to cause equipment and personnel accidents.

Consequently, the rib spalling has become an important factor that restricts the popularization of fully mechanized caving mining with large mining height.

Therefore, the risk assessment for deformation and failure of a coal wall in a fully mechanized caving face with a large mining height has important engineering practical significance for controlling the stability of the coal wall and improving production safety in mines.  

By several scientists were investigated the factors affecting, and control techniques for, rib spalling of the coal wall in the working face to good effect.

In this sense, some of the main results are the following>

·         It was established a mechanical model for assessing the stability of a wedge-shaped sliding mass on the coal wall in the fully mechanized caving face with the large mining height;

·         It was analyzed the development and evolution of mining-induced fractures and revealed the mechanical process of failure of the coal wall based on the theory of the sliding line to reveal the mechanical mechanisms of rib spalling of the coal wall;

·         It was established a mechanical model of sliding and rib spalling of the coal wall based on Bishop’s approach and took the safety factor of sliding plane as an index to judge the stability of coal mass;

·         It was analyzed theoretically the failure modes and forms of rib spalling of the coal wall with a large dip angle and discussed disaster-inducing mechanisms of the surrounding rock by rib-spalling of the coal wall by combining with the spatial structure and movement characteristics of the working face;

·         It was evaluated the stress path effects of rib spalling of the coal wall and built a fracturing-sliding mechanical model for rib spalling of the coal wall in hard, thick coal seams to obtain the relationship between the failure depth and width of the coal wall and the mechanical parameters of the coal mass;

·         In addition, it was established a continuum model based on the progressive S-shaped yield criterion to explore the displacement and stress distribution laws in the coal wall and to predict the extent and rate of cumulative damage of the coal wall;

·         It was established the support and surrounding rock models of a working face with a large mining height under the main roof with different structures and assessed support–roof interactions;

·         Finally, the degree of rib spalling of the coal wall was reduced by optimizing Malisan grouting parameters. By studying influence factors for failure of the coal wall of the fully mechanized face in ultra-thick coal seams, it was identified the relationship between the strength of a coal seam and the thickness of the top coal under different cutting heights.

In previous studies, a coal wall has mostly been taken as an independent research object to explore its properties. However, the instability caused by rib spalling of the coal wall in the top-coal caving face is the result of the comprehensive effects of multiple factors and it is necessary to consider influences of multiple factors, such as mining parameters, roof (top coal) and support on the stability of the coal wall.

The authors established a load-bearing mechanical model of roof–coal wall–support system, based on the moment balance relationship.

Through the theory of elastic mechanics, they obtained the expressions of deformation and stress distribution in the coal wall.

The horizontal displacement of the coal wall was analyzed by taking a top-coal caving face of a given mine as an example.

Furthermore, by using an orthogonal test design and considering different levels of each factor, the horizontal displacement of the coal wall and the volume of a plastic zone were obtained through numerical simulation.

On this basis, the influences of factors affecting the failure of the coal wall and their rank-order according to sensitivity were obtained.

Finally, by combining this with the engineering conditions prevailing in the given mine, the key parameters for controlling and measures for preventing rib spalling of the coal wall were proposed.

They determined three categories of factors influencing the failure of the coal wall. The factors that could be controlled by human actions were the cutting height (or height of top coal), the breaking position of the main roof, the length of top beam of the support, support strength, and the sidewall protecting force. The changes in the horizontal displacement of the coal wall under the influences of different parameters of each factor were assessed.

The order of each factor in influencing the horizontal displacement of the coal wall and the volume of the plastic zone was determined through range analysis. The breaking position of the main roof plays a leading role in the horizontal displacement, followed by the cutting height. The cutting height plays a dominant role in affecting the volume of the plastic zone, followed by the breaking position of the main roof. The support strength and sidewall protecting force have limited effects on the horizontal displacement of the coal wall and the volume of the plastic zone.

The significance of the influence of factors on the horizontal displacement of the coal wall and the volume of the plastic zone was determined through the analysis of variance. In terms of the horizontal displacement, the breaking position of the main roof has the most significant effect, while the cutting height exerts a more moderate influence. As for the volume of the plastic zone, the cutting height and breaking position of the main roof exert highly significant influences and the significance of the cutting height is greater than that of the breaking position of the main roof (the strength of the support exerts a more moderate, general effect).

Technical measures, such as reasonably controlling the cutting height of the working face, increasing the working resistance and sidewall protecting force of the support, and strengthening the management of the coal wall during weighting were proposed. The rib spalling of the coal wall in the given/analyzed working face was effectively controlled, allowing safe, high-efficiency mining of the fully mechanized caving face over its large mining height.

       The obtained performances are very promising and I express my hope to continue their very interesting and useful research.

5) In my opinion, the presented conclusions are suitable related to their research results and prove that they reached the proposed goal.

6) The references in my opinion are very appropriate and their number underlines the scrupulosity of the authors.

7) In this paper, the graphical illustration is well conceived and realized and consequently they contribute to a better understanding of the performed theoretical investigations as well as to underlining the usefulness of the simulation results for the described approach.

I encourage publishing in a new contribution their further results.

Author Response

Point 1-7

Response 1-7: Thank you for your careful reading and analysis of our manuscript.

Our manuscript is based on the engineering background of poor coal wall stability of top coal caving face with large mining height. To prevent serious rib spalling accidents, and to improve the prediction and control capacity for rib spalling, a load-bearing mechanical model of the roof–coal wall–support system was established based on the moment balance relationship, the expressions for the deformation and stress distribution in coal wall were calculated. Then, the influences of key factors on the horizontal displacement of coal wall were investigated. A numerical simulation model of working face was established, and an orthogonal test design was introduced. On this basis, the influences of four factors: cutting height, breaking position of main roof, support strength, and sidewall protecting force of support on the horizontal displacement and volume of a plastic zone of coal wall were analyzed. Moreover, their order of importance was ranked by sensitivity. Based on engineering conditions and production practice of a given Mine, key parameters for controlling and measures for preventing rib spalling of the coal wall were proposed.

Thank you again for your approval of our manuscript. We will conduct further research and strive to publish new research results as soon as possible.

Special thanks to you for your comments again. They are valuable and very helpful for revising and improving our paper, as well as the important guiding significance to our researches.

Reviewer 2 Report

Overall, the study presents a sound theoretical argument for assessing the risk of coal wall deformation and failure. The approach adopted to improve the prediction and control capacity for rib spalling through determining the load-bearing mechanical model of the roof–coal wall–support system by calculating the moment balance relationship, the expressions for the deformation and stress distribution in the coal wall.

However, more could have been done to highlight the study limitations and suggest potential recommendations for future studies to build on the findings emanating from this study. 

Author Response

Point 1: However, more could have been done to highlight the study limitations and suggest potential recommendations for future studies to build on the findings emanating from this study.

Response 1: Thank you for your suggestion.

As we all know, no one's research can be perfect, and this manuscript is no exception. The manuscript puts forward a bearing mechanics model that comprehensively considers coal seam occurrence influencing factors, internal influencing factors and external influencing factors, uses elastic mechanics theory to solve the deformation and stress distribution expression in coal wall, explores the influencing factors and rules of the horizontal displacement of coal wall, and obtains the horizontal displacement and plastic zone volume of coal wall through orthogonal test method and numerical simulation method. The influence rule and sensitivity ranking of coal wall failure factors are carried out, and finally, based on the engineering practice conditions, the key parameters for controlling and measures for preventing rib spalling of the coal wall were proposed.

In this manuscript, in the process of establishing the mechanical model, the geological conditions were simplified and assumed, such as the coal seam is homogeneous, do not consider the hinge relationship between rock blocks, etc., similarly, in the selection of influencing factors, only the theoretical model used to analyze the factors, while excluding some human controllable factors, these are the limitations of this study. These are also potential directions for future research.

According to the findings of this study, the cutting height and breaking position of the main roof were found to significantly affect rib spalling of the coal wall, while the support strength and sidewall protecting force only slightly influence rib spalling. It is an important engineering requirement to control the mining height of the working face and maintain the coal wall after the roof is broken. In the future, the influence of mining height on roof breaking position can be studied, and the interaction relationship between mining height, roof breaking position and coal wall wall can be analyzed, so as to improve the prediction and control ability of rib spalling more accurately.

Special thanks to you for your comments again. They are valuable and very helpful for revising and improving our paper, as well as the important guiding significance to our researches.

Reviewer 3 Report

The article can be accepted for publication, taking into account the elimination of these shortcomings.

1.     The main shortage of the work is the inconsistency of the title with the results of the work. The word risk is mentioned on pages 10, 19, 34, 200, 365. But the risk is not considered and evaluated as a probabilistic quantitative measure of the danger of bottomhole collapses. The words risk should be removed from the title of the article.

2.     The second shortage is that the influence of the heterogeneity of the structure and properties of the coal seam is not taken into account. This can be taken into account by the coal seam damage factor.

3.     Usually, to determine the safe conditions of a coal face, a stability factor is determined. In this paper could give estimates of stability factor and study the influence of the factors under consideration on this coefficient.

4.     The purpose of the article is recommended to highlight and justify in the introduction.

Author Response

Point 1: The main shortage of the work is the inconsistency of the title with the results of the work. The word risk is mentioned on pages 10, 19, 34, 200, 365. But the risk is not considered and evaluated as a probabilistic quantitative measure of the danger of bottomhole collapses. The words risk should be removed from the title of the article.

Response 1: Thank you for pointing this out. In recognition of this, we removed the word ‘risk’ from the title and amended it appropriately. The relevant sentences have been modified in the revised manuscript.

Point 2: The second shortage is that the influence of the heterogeneity of the structure and properties of the coal seam is not taken into account. This can be taken into account by the coal seam damage factor.

Response 2: Thank you for pointing this out. As you said, the coal seam is indeed heterogeneous, with cracks of different sizes and directions. Before mining, coal seam is in a complete state, and its mechanical properties can be expressed by elastic modulus E. During mining, the coal mass in a part of area in front of the coal wall has yielded, making this a problem of plastic mechanics. By reducing the elastic modulus of coal mass, it is transformed into a problem of elastic mechanics for approximate solution. The relevant explanations have been added to the revised manuscript.

The failure coefficient of coal seam you proposed is more like the description of the failure degree of coal seam rather than the description of the heterogeneity of coal seam. Similarly, the heterogeneity of coal seam can also be simply summarized by increasing and decreasing the elastic modulus E.

Point 3: Usually, to determine the safe conditions of a coal face, a stability factor is determined. In this paper could give estimates of stability factor and study the influence of the factors under consideration on this coefficient.

Response 3: Thank you for pointing this out. The term ‘stability factor’ generally appears in open-pit mining slope, which mainly refers to the sliding failure safety of slope. Meanwhile, in the papers of Yuan (Yuan et al. 2013) and Li (Li et al. 2015), stability factor is also used to evaluate the safety of coal wall rib spalling. The common point is that the models are all sliding models, and the safety factor of sliding surface is taken as the evaluation index.

However, this manuscript adopts the evaluation method of displacement and plastic zone, and some scholars also evaluate the stability of coal wall by calculating and monitoring displacement and plastic zone (Kong et al. 2021, Rashed et al. 2021 and Guo et al. 2019). Plastic zone indicates that the coal seam changes from elastic state to plastic failure, and the bearing capacity decreases greatly, which can represent the maximum depth of coal wall. The horizontal displacement indicates the deformation degree of coal wall. The greater the horizontal displacement, the more severe the deformation of coal wall and the possibility of crushing.

The authors believe that these are two different ways of measuring the failure of coal wall, and the revised title is more consistent with the evaluation method adopted in this paper.

In the future study, we will build slip model of coal wall rib spalling to explore the factors affecting the stability factor of sliding surface and the change of the stability factor of sliding surface in the failure process.

Point 4: The purpose of the article is recommended to highlight and justify in the introduction.

Response 4: Thank you for pointing this out. Due to the large mining height of top coal caving in Cuncaota II Coal Mine, the mechanical mining height increases, which leads to the instability problem of coal wall rib spalling, resulting in serious safety problems. The analysis of coal wall deformation and failure factors is carried out, and the key controlling factors are pointed out by studying the influence law and sensitivity ranking of each factor controlling coal wall rib spalling, and the prevention and controlling measures are formulated accordingly, which is helpful to reduce the occurrence of rib spalling accidents of working face and ensure the safe and efficient production of working face. The relevant explanations have been added to the revised manuscript.

Yuan, Y., Tu, S.H., Zhang, X.G. (2013), “Mechanism and control technique of rib spalling disaster in fully-mechanized mining with large mining height in soft coal seam face”, Disaster Advances, 6:92-98.

Li, X.P., Kang, T.H., Yang, Y.K.(2015), “Analysis of coal wall slip risk and caving depth based on Bishop method”, Journal of China Coal Society, 40(7):1498-1504. https://doi.rog/10.13225/j.cnki.jccs.2014.0912.

Kong DZ, Xiong Y, Cheng ZB, Wang N, Wu GY, Liu Y (2021) Stability analysis of coal face based on coal face-support-roof system in steeply inclined coal seam. Geomech Eng 25: 233-243. http://dx.doi.org/10.12989/gae.2021.25.3.233

Rashed G, Mohamed K, Kimutis R (2021) A coal rib monitoring study in a room-and-pillar retreat mine. Int J Min Sci Technol 31: 127-135. https://doi.org/10.1016/j.ijmst.2020.10.001

Guo WB, Liu CY, Dong GW (2019) Analytical study to estimate rib spalling extent and support requirements in thick seam mining. Arabian J Geosci 12. https://doi.rog/10.1007/s12517-019-4443-8

Special thanks to you for your comments again. They are valuable and very helpful for revising and improving our paper, as well as the important guiding significance to our researches.